
# Identification of platform exhaust on the RV Investigator

Ruhi S. Humphries[1], Ian M. McRobert[2], Will A. Ponsonby[2], Jason P. Ward[1], Melita D. Keywood[1], Zoe Loh[1], Paul B. Krummel[1], and James Harnwell[1]

[1]Climate Science Centre, CSIRO Oceans and Atmosphere, Aspendale, Australia
[2]Engineering and Technology Program, CSIRO Oceans and Atmosphere, Hobart, Australia

*Correspondence to:* Ruhi Humphries (Ruhi.Humphries@csiro.au)

**Abstract.** Ship-based measurements are an important component in developing an understanding of the global atmosphere. A common problem that impacts the quality of atmospheric data collected from marine research vessels is exhaust from both diesel combustion and waste incineration from the ship itself. Described here is an algorithm, developed for the recently commissioned Australian blue-water Research Vessel (RV) Investigator, that identifies exhaust periods in sampled air. The RV

Investigator, with two dedicated atmospheric laboratories, represents an unprecedented opportunity for high quality measurements of the marine atmosphere. The algorithm avoids using ancillary data such as wind speed and direction, and instead utilises components of the exhaust itself - aerosol number concentration, black carbon concentration, and carbon monoxide and carbon dioxide mixing ratios. The exhaust signal is identified within each of these parameters individually before they are combined and an additional window filter is applied. The algorithm relies heavily on statistical methods, rather than set-

ting thresholds that are too rigid to accommodate potential temporal changes. The algorithm is more effective than traditional wind-based filters in removing exhaust data without removing exhaust-free data which commonly occurs with traditional filters. With suitable testing, the algorithm has the potential to be applied to other ship-based atmospheric measurements where suitable measurements exist.

## 1 Introduction

When undertaking atmospheric composition and chemistry measurements, a common issue that impacts data quality is the ability to effectively identify and potentially filter out sources of contamination. The most common local contamination source is often emissions from power generation. Typically, power generation burns hydrocarbon fuels (such as diesel) and emits a range of combustion products that are often the target species being measured in the background atmosphere.

Identification of periods of contamination is performed via a variety of methods depending on the contamination source and the target research question. A commonly used and reasonably reliable method for identification of local point source contaminants is by simple wind direction and speed criteria (e.g., Molloy and Galbally, 2014; Steele et al., 2003; Chambers et al., 2017, and references therein). This method aims to capture the exhaust plume diffusion processes using the two wind





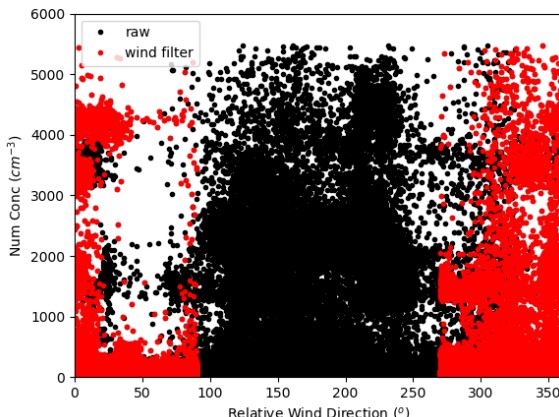

**Figure 1.** CCN plotted against wind direction (relative to the platform) from the RV Investigator voyage IN2016_V03. Black: unfiltered data. Red: after data is removed when wind speed is less than $5 \text{ m.s}^{-1}$ or between $90^{\circ}$ and $270^{\circ}$. Uncontaminated data are usually less than $1000 \text{ cm}^{-3}$. Data filtered with just wind measurements still show clear signs of contamination.

measurements as proxies. It is a robust method in environments where background composition is similar to the contamination source, such as in urban areas. However, because of the oversimplified parameterisation, very conservative bounds are often required which results in the removal of often significant amounts of contaminant-free data. In addition, this method assumes relatively uniform flow characteristics and will fail when atmospheric recirculation results in measurements of contaminated air

from directions outside the specified range. Figure 1 exemplifies this issue, where Cloud Condensation Nuclei (CCN) number concentrations are found to be unreasonably high for the marine dataset used here, even after a wind speed and direction filter is utilised.

Depending on the environment, a combination of wind criteria and *in-situ* composition measurements can be used to help overcome the recirculation issue. For example, high concentrations of nitrogen oxides ($NO_x$) produced from combustion pro-

cesses will react rapidly with background ozone ($O_3$), resulting in $O_3$-depleted air which will only regenerate hours downwind through $NO_x$ chemistry and photolysis processes (World Meteorological Organization (WMO), 1985). The use of $O_3$ can improve wind-based filters to help identify recirculation, depending on the time-scale of interest (e.g. Humphries et al., 2015). However, the problem of false-positive identification remains as long as measurements of ancillary data are used for identification. Ideally, identification of contaminated air would use only measurements of species emitted directly by the source itself

in order to minimise false-positive contaminant identification and maximise the usable data from a dataset.

In the current study, an exhaust identification algorithm is developed for application to data collected on-board Australia's new marine Research Vessel (RV) Investigator utilising measurements of species emitted directly by combustion processes occurring on the ship - namely diesel combustion and waste incineration. Both combustion processes (hereafter referred to as 'exhaust') have similar emissions relative to the background atmosphere (Reşitoğlu et al., 2015; Johnke, 1999; Jones and

Harrison, 2016, and references therein). Emitted species include carbon dioxide ($CO_2$), carbon monoxide (CO), $NO_x$, hy-



drocarbons, as well as high concentrations of aerosols (Condensation Nuclei, CN) which include those whose composition is primarily black carbon (BC) as well as those that can act as CCN. Measurements of CO, $CO_2$, BC and CN are utilised for the development of this exhaust identification algorithm as they have clear signals above the background atmosphere and are measured routinely on the vessel.

Figure 2 shows an example period of data from the vessel that illustrates the different signals resulting from exhaust influence that must be characterised in the algorithm. Exhaust influence in CN, CO and $CO_2$ data is obvious with striking enhancements above variable background signals throughout the sample period. BC data are generally close to zero, with exhaust influence obvious when a signal appears out of the noise. Strong perturbations over extended periods, such as those observed on May 18, are indicative of direct exhaust influence. Smaller signals, such as those observed in CN data on the afternoon of May 19,

or in BC data on May 20, indicate a more dilute influence, with sampling likely occurring on the wavering edge of the exhaust plume.

    Not all measured parameters respond to the exhaust simultaneously, or necessarily at all. This is most obvious in the extended exhaust event that occurred around 0000 hours on May 19 (Figure 2) where a strong signal is observed in CN data, but is absent in the other species. This is likely a result of the differences associated with the measurement techniques of the different species.

Being a simple counting instrument, the CPC is sensitive to particle concentrations down to 1 $cm^{-3}$. For the CO and $CO_2$ measurements, although the precisions are high, the flow through cell technique utilised results in physical integration of the sample over a minute, thereby smoothing out any perturbations. For BC measurements, the detection limit of the instrument is 50 $ng.m^{-3}$ over a 10 minute average. At 1 Hz time resolution, we are still able to get useful signal (for the current purpose) at 0.1 $ng.m^{-3}$ mass resolution, however the instrument is clearly missing significant exhaust influence. In addition, since the

sizes of exhaust particles is sub-100 nm (Humphries et al., 2018b), the mass-based measurement of the BC is significantly less sensitive to it compared to the number-based CPC measurements. Lastly, differences in residence times in the instruments can result in time offsets in the signals between instruments.

    The RV Investigator is a blue-water research vessel capable of traversing from the ice-edge to the equator. The types of atmospheres it encounters range from pristine background, to continental (e.g. while sampling near the coast), to urban envi-

25 ronments (e.g. while in port). An important objective of this algorithm is the ability to distinguish the local ship exhaust from the atmosphere of interest - a task which becomes particularly difficult in the more polluted environments such as those downwind of large urban centres. In this study, the dataset utilised for development contains influences from urban and background marine regions (as shown in Figures A2 and A3) by which differentiation from ship exhaust can be achieved.

    In this study, an algorithm is developed that produces an exhaust identification product that is published alongside other

publically available datasets from this platform. The algorithm aims to accurately identify exhaust from the ship itself, distinct from other polluted atmospheres such as urban centres, and minimise false-positive identification in order to retain as much valuable data from this mobile platform as possible. The exhaust product is developed utilising a dataset exemplifying the range of atmospheres that are sampled and is validated by applying it to measurements of CCN that were measured simultaneously.





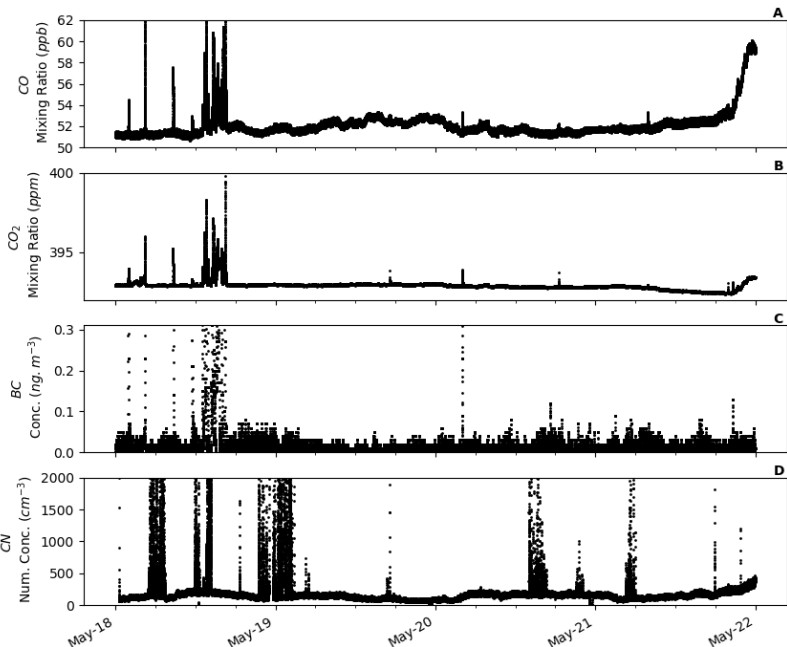

**Figure 2.** A five day subset of the data from 2016 used to illustrate the exhaust algorithm throughout the paper (full data shown in Figures A2 and A3). All data are raw instrument output without calibrations to aid in rapid dissemination of the exhaust identification product. Timestamps are UTC. Note that y-axes are limited in range to reveal baseline values. Exhaust signal for CO, $CO_2$, BC and CN extends up to 800 ppb, 490 (ppm), 10 ng.m$^{-3}$ and $10^6$ cm$^{-3}$, respectively.

## 2 Instrumentation

The RV Investigator is a state-of-the-art research platform commissioned in 2015 by the Australian government. The vessel is designed for blue-water research and is capable of spending up to 300 days per year at sea, with a single voyage up to 60 days and over 10000 nautical miles. Propulsion and power is provided by two diesel-electric engines together with three 3000 kW, nine cylinder diesel engines. Exhaust from diesel combustion, together with waste incineration which is emitted from a separate but co-located flue, provides the largest source of contamination to atmospheric measurements aboard the platform.

The vessel has been purpose built with two dedicated atmospheric laboratories along with a custom-designed air sampling inlet located above the ship's bow, approximately 18.4 m above sea level. The aerosol laboratory is situated directly underneath the air sampling inlet fore of the anchor well, such that the distance between sampling and instrumentation, and thus sample





losses, are minimised (total distance to the aerosol laboratory's sampling manifold is ∼8 m). The aerosol laboratory houses instrumentation for the measurement of aerosols and reactive gases such as ozone and nitrogen oxides. The air chemistry laboratory is situated further aft in the vessel at the fore of the superstructure (total distance from main sample inlet to the air chemistry laboratory's sampling manifold is ∼38 m), and houses instrumentation for the measurement of less reactive

atmospheric species such as greenhouse gases and volatile organic compounds.

The RV Investigator houses a range of permanent instrumentation. These instruments are run continuously throughout every voyage of the RV Investigator (except for when instruments are removed for maintenance or faults) and after data have been calibrated, and quality assurance and control procedures have been performed, data are made publicly available. Of particular relevance to this study is the measurement of CO, $CO_2$, BC, CN, and meteorological measurements. Each of these parameters

will be described in detail in future publications documenting the ongoing measurements of the vessel, however a brief overview of these measurements are given here. For this analysis, CCN data are utilised as an independent parameter by which the exhaust identifier is tested. The dataset considered in this manuscript utilised CN and CCN data captured by instrumentation deployed specifically for this voyage, and thus will be described separately. It is worth noting that both CN and CCN instrumentation have more recently become part of the permanent ongoing instrument suite and will be described in a future publication and

made publicly available alongside other aerosol data from the platform.

An important outcome of the current work is to make publicly available an exhaust identification data product that will be published alongside other atmospheric datasets from the vessel in order to assist data users in their analyses. For the present manuscript, the exhaust identification product has been developed using data from the RV Investigator voyage IN2016_V03 (see Marine National Facility (2016) for voyage track), and data utilised and produced in this manuscript is available from

20 Humphries et al. (2018a).

## 2.1 Carbon monoxide, CO

Mixing ratios of carbon monoxide (CO) were measured continuously at 1 Hz using a mid-infrared (IR) quantum cascade laser spectrometer (Aerodyne Research Inc, Billerica, MA, USA). A high vacuum dry scroll pump (Model SH-110, Varian, Lexington, MA, USA) draws air through the 0.5 L optical cell maintained at a constant pressure of 45 Torr, and flushed at a

25 rate of approximately 0.5 L.min$^{-1}$. Mid-IR laser light enters the astigmatic multi-pass cell, traversing it 238 times, giving an effective path length of 76 m. Upon exit from the optical cell, the light impinges on a thermoelectrically cooled IR detector, allowing a mixing ratio to be determined via Beer's Law. The nominal precision of the CO measurement is 60 ppt in one second (owing to the long path length and strong transition of the CO molecule in the mid-IR). Water vapour is also measured allowing for the CO mixing ratio to be corrected to a dry air mixing ratio, without the need to pre-dry the sample.

## 30 2.2 Carbon dioxide, CO$_2$

Atmospheric mixing ratios of carbon dioxide ($CO_2$) are measured on board the RV Investigator using a Picarro cavity ringdown spectrometer (Model G2301, unit CFADS2315, Picarro Inc., Santa Clara, CA, USA), that concurrently measures methane ($CH_4$) and water vapour. Air is drawn through the 35 sccm optical cell held at constant temperature and pressure (45ºC and



140 Torr), at a rate of approximately 0.15 $L.min^{-1}$. The ends of the cell comprise highly reflective mirrors that recirculate the light supplied by a near-infrared (NIR) laser through the cavity, resulting in an effective path length of around 20 km. Light leaks out of the mirrors, impinging on a photodetector with a characteristic ringdown time. Carbon dioxide molecules within the cell also absorb a fraction of the light, modulating the ringdown time in proportion to their concentration. By scanning the

5   laser off the absorption peak and remeasuring the ringdown time, the technique becomes insensitive to fluctuations in laser power. The precision of the $CO_2$ measurement is better than 0.05 ppm on a minutely average. Data used in this paper are raw $CO_2$ dry air mixing ratios (by an empirical correction using the native water vapour measurement). $CO_2$ data are also available as minutely and hourly mean dry air mixing ratios that have been calibrated and drift corrected through the daily measurement of a reference tank.

## 10   2.3   Black carbon, BC

Black carbon measurements are made using a Multiangle Absorption Photometer (MAAP Model 5012, Thermo Fisher Scientific, Air Quality Instruments, Franklin, MA, USA). The MAAP collects aerosol on a glass fibre tape that gets irradiated with 670 nm light. Photo-detectors measure the light transmission and reflection in the forward and back hemispheres, respectively, and after inversion, reports black carbon concentrations in real time. The inversion algorithm takes into account multiple scat-

tering processes inside the aerosol sample and between the sample and the filter matrix and utilises a carbon mass absorption coefficient of 6.6 $m^2.g^{-1}$. The detection limit of the instrument over 10 and 20 min is 50 and 20 $ng.m^{-3}$, respectively.

## 2.4   Aerosol number concentration, CN

Number concentrations of condensation nuclei larger than 3 nm (CN) were measured continuously at 1 Hz using a condensation particle counter (CPC Model 3776, TSI, Shoreview, MN, USA). The CPC works by drawing the aerosol sample continuously

through a chamber of supersaturated 1-butanol which condenses onto particles larger than 3 nm, growing them to sizes (above 1 μm) which can be counted individually by a simple optical particle counter. Sample flow rate is regulated by a critical orifice at 1.5 $L.min^{-1}$. This flow rate was checked every few days at the instrument inlet using an external flow meter (Sensidyne Gilibrator, St. Petersburg, FL, USA) and flow rates were found not to deviate beyond 1%. Although flow calibrations weren't necessary for this algorithm, the software used for filtering the data simultaneously performs flow calibrations, so calibrated

data is used here. Data are also filtered for periods of instrument zeros and the disconnection of the instrument from the sampling line. Note that for voyages after September 2016, a permanent CPC (Model 3772, TSI, Shoerview, MN, USA), measuring CN larger than 10 (nm), was installed on the platform (described in detail in future publications) and is used as the CN data stream.

## 2.5   Cloud Condensation Nuclei, CCN

Number concentrations of cloud condensation nuclei (CCN) were measured continuously using a continuous-flow streamwise thermal-gradient CCN counter (CCNC, Model CCN-100, Droplet Measurement Technologies, Longmont, CO, USA).



The instrument was situated at approximately the same distance from the inlet as the CPC, connected to the manifold using a combination of stainless steel and flexible conductive tubing. The instrument was configured to run continuously at 0.5% supersaturation, which after pressure calibrations, was found to equate to 0.5504% supersaturation. The flow rate of the instrument was set to the standard 0.5 L.min$^{-1}$. Flows were checked weekly using an external flow meter (Sensidyne Gilibrator,

St. Petersburg, FL, USA) and concentrations were corrected in post-processing procedures based on actual flow rates (maximum of 2% flow deviation). Data were quality controlled by removal of periods during which maintenance was performed and calibrated for pressure and flow rates.

## 2.6 Meteorological data

Meteorological data were measured continuously whilst the ship was underway. Meteorological measurements include air
temperature, relative humidity, barometric pressure, solar radiation, precipitation, sea surface temperature, wind speed and direction. Of particular interest to the exhaust filtering algorithm are measurements of wind speed and direction. Dual wind monitors (Marine Wind Monitor, Model 05106, R.M. Young Company, Traverse City, Michigan, USA) are affixed to the vessels foremast at a height of 24 m from the water line, each offset from the ship's centreline by ∼2.5 m, one to starboard and the other to port. The measurable wind speed range of the wind monitors is 0-100 m.s$^{-1}$ (±1%), with an azimuth range of 0-355° (±3°;
relative to ship centre line; the 5° dead-zone of which is directed aft). An ultrasonic 2-axis anemometer (WindObserver II, Gill Instruments, Lymington, Hampshire, UK) is also affixed to the foremast 21 m from the water line and ∼2.5 m to port from the ship's centreline. The ultrasonic anemometer measures wind speed in the range 0-65 m.s$^{-1}$ (0.01 m.s$^{-1}$ resolution and ±2% at 12 m.s$^{-1}$) and azimuth range of 0-359° (1° resolution and ±2% at 12 m.s$^{-1}$). Wind sensors are calibrated annually by Ecotech Australia to the reference standard ISO 17713-1:2007.

## 3 Exhaust Identification

The primary task of the algorithm is that of distinguishing between two distinctly different signals in our data. Because of the magnitude of the difference, a first pass of the exhaust identification is simply an application of outlier detection algorithms. However, on closer inspection, the variability of the exhaust signal due to variations in source strength, dilution and plume location sampling, as well as the shear length of time that the exhaust can influence measurements (from seconds to days)
makes many of the more well-known detection algorithms unsuitable to this problem. This is discussed more in Appendix Section A where a number of algorithms, including Fast Fourier Transform, z-score and modified z-score, double exponential smoothing and histogram methods, were tested and found to be unsuitable. Hence this complicates the goal of the algorithm to differentiate between these two distinct but varying signals (i.e. exhaust and ambient in a range of environments).

Exhaust identification is performed primarily utilising the intersection between four parameters commonly emitted in fossil
fuel combustion processes, namely CO, $CO_2$, BC and CN. Figure 2 shows the variability of these species during periods of exhaust influence and within background air (defined here as not influenced by exhaust from the measurement platform, the





RV Investigator). Distinct signals are observed in all four variables, however it is important to note that not all signals respond simultaneously. This concept is discussed in detail later in the manuscript.

Because of the differences in their exhaust responses, identification is performed on each of the parameters separately, after which they are combined and an additional window filter is applied to remove neighbouring values that are not captured

completely by the parameters themselves. Each instrument connected to the Investigator's sampling system will also exhibit temporal variations in their responses to exhaust strikes due to differences in residence and detector response times. Because of this, it is impossible to create a single exhaust identification product that can be applied to every instrument that collects data on this platform. To effectively achieve a perfectly exhaust free dataset for each instrument without removing substantial data that is free from exhaust, identification should ideally be performed on each dataset individually. Nevertheless, the creation of this

exhaust identifier product is useful in that it creates a first-pass filter that identifies the vast majority of the exhaust influence. With this in mind, a relatively conservative approach is adopted in order to strike a balance between not identifying periods of exhaust influence, and the false-positive identification of background data as exhaust. Since the product is not used to filter published datasets, but instead is published alongside other data, it is left to the end-user to determine whether more stringent criteria should be applied to specific data sets than the relatively conservative approach adopted here.

**3.1   BC threshold filter**

In the background atmosphere, BC is generated from combustion sources such as fossil fuel burning and biomass burning (Seinfeld, J.H., Pandis, 2016). Moreover, the lifetime of BC is on the order of days (Cape et al., 2012) and combined with transport dilution, seeing elevated values beyond the instrument sensitivity is rare. This is illustrated in Figure A3 where the baseline trend observed in CN during a period of urban influence (May 26) is absent in the BC data. Consequently, a set

threshold value can be utilised for BC, whereby any data above this threshold are identified as exhaust.

The threshold for exhaust was determined by selecting numerous periods where background air was being measured, and selecting the maximum value during these periods. For the dataset being utilised for this manuscript, a value of $0.07\ \mu g.m^{-3}$ was chosen. This value is suitable for those measurements made when significant biomass burning events are absent, however alteration of this value is possible for voyages where this influence is significant.

**3.2   Variance filter for CO, CO$_2$ and CN**

In contrast to BC, CO, CO$_2$ and CN all have persistent, non-zero background signals in the atmosphere and consequently, a simple threshold filter cannot be utilised. For these datasets, the variability is characterised on each dataset and outliers in the positive direction are identified as exhaust. As discussed by Leys et al. (2013), the robust statistical parameters of median and median absolute deviation (MAD) are useful in the detection of outliers since they are relatively insensitive to outliers

compared to the mean and standard deviation (SD) that are commonly utilised.

For a univariate dataset $x_1, x_2, ..., x_n$, the MAD is defined as the median of the absolute deviations from the data's median:

$$MAD = median(|x_i - median(x)|) \tag{1}$$




It is well established that for normally distributed data (such as is being explored here for data without exhaust), the median and mean are equivalent. The same can be said for the MAD and the SD provided a standard factor is applied (Rousseeuw and Croux, 1993, and references therein) such that:

$$SD = 1.4826 \times MAD \tag{2}$$

To identify the exhaust, the data point in question must be assessed to determine if it is within an acceptable range that represents the background atmosphere. Defining this acceptable range deserves thoughtful consideration. Given the variability of $CO$, $CO_2$ and CN in the background atmosphere, a predefined range would not be fit-for-purpose. This circumstance lends itself naturally to the use of a rolling window. For this algorithm, numerous statistical parameters (median, MAD and SD) are calculated on a detrended, centred rolling five minute window. Although variable, the five minute width of this rolling window

is chosen here so that there is enough data for statistical robustness, yet short enough to capture real changes in atmospheric state.

It is important to note that when the fraction of outliers dominates ($50\%$) a sample (or window), median based statistics also become sensitive to outliers. This will happen when, for example, the rolling window is sampling during an exhaust period that persists longer than half the window period. To get a statistical dataset that represents the background atmosphere to which raw

data can be compared, alternative values must be sought during these periods when all calculated statistics are affected.

The first step in this process is to identify periods where median based statistics are affected in the rolling window. Comparing the rolling SD ($SD_i$) and MAD ($MAD_i$) could be effective for identifying these periods, since one is sensitive to outliers while the other is not, respectively. However, since the exhaust could represent up to 100% of the sample window, the rolling MAD ($MAD_i$) and SD ($SD_i$) could be similar, ruling out comparing these two parameters as a method for identification. To overcome

this, a single MAD value ($MAD_B$) that is representative of the background atmosphere is sought to which we can compare $SD_i$.

Analysis of both CN, CO and $CO_2$ data show that $MAD_i$ are generally tightly grouped, but have a small fraction of large outliers, as shown in Figures A4, A5 and A6. Choosing the median of this $MAD_i$ dataset, $MAD_B$, yields the value representative of the background atmosphere to which $SD_i$ can be compared and exhaust affected median statistics can be replaced. Time

periods with $SD_i$ larger than three times $MAD_B$ are then flagged and values during these periods are replaced with values obtained by linear interpolation with neighbouring values, yielding new datasets, $M_i^B$ and $MAD_i^B$ that represent the rolling median and MAD without influence from exhaust. Having obtained statistical datasets reasonably free from exhaust influence, exhaust can be identified in the raw data such that:

$$x > M_i^B + 3MAD_i^B \tag{3}$$

where $x$ is the raw CO, $CO_2$ or CN data.

The algorithm only identifies positive deviations as exhaust, ignoring negative outliers. This is done because the exhaust can only add signal to the background for these three parameters at this range and at this high frequency. Inclusion of the lower limit could erroneously identify exhaust time periods which are simply instrument zeros or calibrations that may not have been removed from the datasets prior to their use in the algorithm.





The use of uncalibrated and uncorrected CO, $CO_2$ and CN data is acceptable within the algorithm so long as periods of instrument calibrations in the positive direction are removed from the datasets prior to use (only positive since the exhaust influence on these parameters are all in this direction). This is because the algorithm is sensitive to high-frequency changes like exhaust strikes or instrument zeros, rather than lower frequency variations, such as instrument drifts, and takes no account for
the absolute value of the signal.

### 3.3   Window filter

Once identified by either CO, $CO_2$, BC or CN, a window filter is applied. This rolling filter sums the number of exhaust points in the window. If this sum is larger than 10% of number of points in the window, then all data points within that window are labelled as exhaust. The 10% threshold is important because variations in one of the three parameters (arising from the
use of raw data streams) could mistakenly identify a time period as exhaust without verification from either sustained exhaust identification, or other parameters. Additionally, this 10% threshold, together with the choice of the window width (here set to 20 minutes), creates a buffer that accounts for differences in residence times of atmospheric samples in the sample lines and in the instruments themselves.

### 4   Results and Discussion

Figure 3 shows a subset of data to illustrate the exhaust filter when applied to the CCN dataset. CCN is used here as an independent dataset to test the exhaust filter algorithm and ensure its applicability beyond the parameters used in the algorithm itself. In addition, exhaust strikes are easily visible in the CCN dataset, making it useful for this purpose.

Although not 100% effective, the algorithm removes the vast majority of exhaust influence and its effectiveness is clearly apparent (Figure 3). It is clear that none of the parameters are capable of entirely capturing the exhaust influence individually,
as the influence of the exhaust is not obvious in all parameters simultaneously. Periods occur, such as on the afternoon of May 18, where only part of the exhaust period is captured in the CN data, while CO, $CO_2$ and BC data show quite strong signals. Conversely, around 0000 hours on May 19, there is no evidence of exhaust signal in the CO, $CO_2$ or BC data, while a strong signal is observed in the CN data.

This is further illustrated in Figure 4 where each parameter of the filter is applied to the CCN dataset separately. Individually,
each filter removes only a small fraction of the exhaust influence. The CN filter is the most effective, presumably because the exhaust signal is so distinct from background values and the response time is rapid. Nevertheless, a significant fraction of exhaust periods make it through the CN filter. When all parameters are used together, the exhaust filter improves dramatically, although a small fraction of exhaust values remains. The application of the window removes most of the remaining exhaust affected data, resulting in a dataset that can be confidently used in subsequent analyses of the background atmosphere.
Figure 5 shows different combinations of the individual filters to demonstrate the effectiveness of each filter. Combining both Figures 4 and 5 indicates that CN is the most effective parameter, followed by BC, CO, then $CO_2$. By itself, CN removes the vast majority of the exhaust influence, but alone is incomplete. While this suggests that a simple filter utilising CN and only one



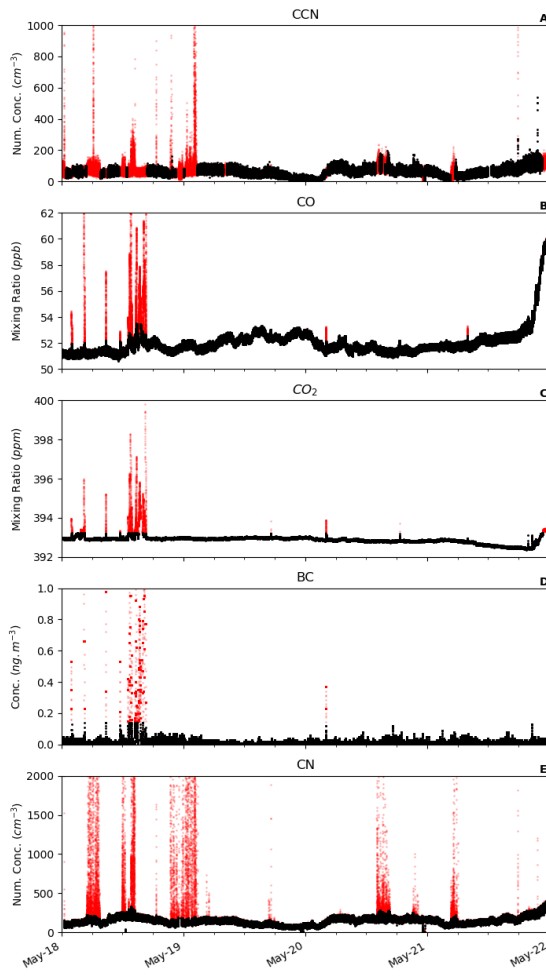

**Figure 3.** A subset of the data (same as shown in Figure 2) with the filtered data (black) shown atop the raw data (red). Panel A shows the unfiltered CCN data along with data after the full filter is applied. Panels B to E show the filter parameters as both raw as well as with their individual filters applied.





of the other three parameters could be used to produce a similarly effective filter, in practice, having all three measurements (i.e. BC, CO and $CO_2$) provides important redundancy. Currently, if problems occur with the CN measurements, the effectiveness of the exhaust filter is significantly reduced, as shown by Figure 5H. Given the importance of the CN data to being able to effectively identify exhaust, instrumental redundancy for CN measurements is an important feature of the platform that should
be considered in the future.

Interestingly, there are some periods which still show short periods of exhaust in the filtered CCN data, as shown in Figure A8. Here, the exhaust is easily identified by the CN filter algorithm, however the exhaust signal is delayed in the CCN data by about 10 s, presumably due to small differences in instrument residence times (or timestamps associated with the campaign based instrument used in this manuscript). It is possible to alter the algorithm in such a way that the 20 min window applies
to any period identified as exhaust (rather than having the threshold described in Section 3.3), however this has the immediate ramification of large losses of data that would otherwise be classified as background, which would be unacceptable for this purpose. Instead, this exhaust identifier has been designed to be used as an initial step, and that if more stringent bounds are required by the end-user, a more strict window filter can be applied at that time. In addition, individual datasets should be analysed for any remaining exhaust to account for differences in residence time and sampling regimes.

Also apparent in Figures 4 and A7 is the advantage of this algorithm compared to the standard wind filter shown in Figure 1. The primary influence of the exhaust is shown in the raw data between relative wind directions of approximately 90° and 270°. Since the algorithm here doesn't use wind parameters, filtered data appear from wind directions where traditionally it would be removed. This represents a significant recovery of data. For this 42 day dataset, data that would have been removed by the wind direction filter, but have remained with this algorithm equate to almost 20 hours of observations recovered. In addition,
the algorithm removes data that would have been entirely missed by a wind filter, as illustrated by Figure 1.

## 5   Conclusions

A ship exhaust identification algorithm is described that utilises only components of the combustion exhaust, rather than commonly utilised ancillary data such as wind speed and direction. CO, $CO_2$, BC and CN data are used as exhaust indicators and together with surrounding window removal, a robust exhaust identification method results. Statistical methods feature
heavily in the algorithm in order to avoid as much as possible, cut-off thresholds that can be subjective. The algorithm is applied directly to data from the RV Investigator for which it was specifically developed and the resulting data product will be made available along-side other publicly available data from the research platform. The algorithm performs well and identifies the vast majority of exhaust in the dataset while minimising data loss that can occur with other overzealous or indiscriminate methods.

*Data availability.* Input data and the exhaust product calculated for the sample data utilised in this manuscript are available at Humphries et al. (2018a). Please contact the author for access to code.





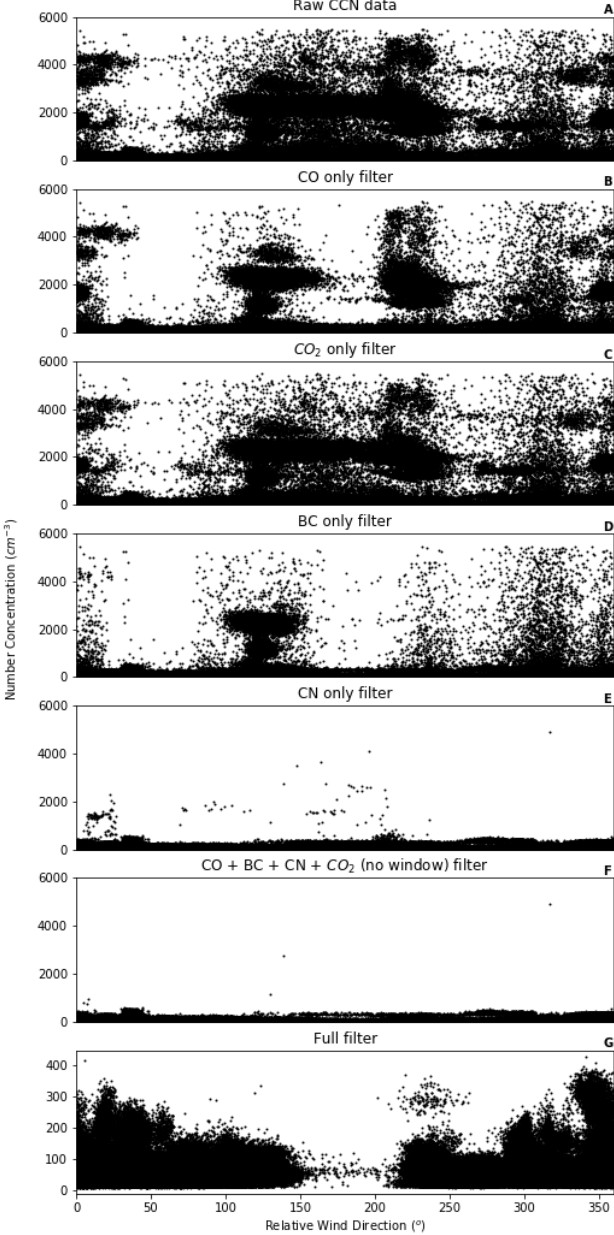

**Figure 4.** CCN data (linear scale) with the different steps of the algorithm applied separately: panel A shows unfiltered data; panels B to E show single parameter filters; panel F show the combination of the four parameters; and panel G shows the full filter which includes all parameters and the application of a window removal. Note the change in scale of the y-axis in the final panel.





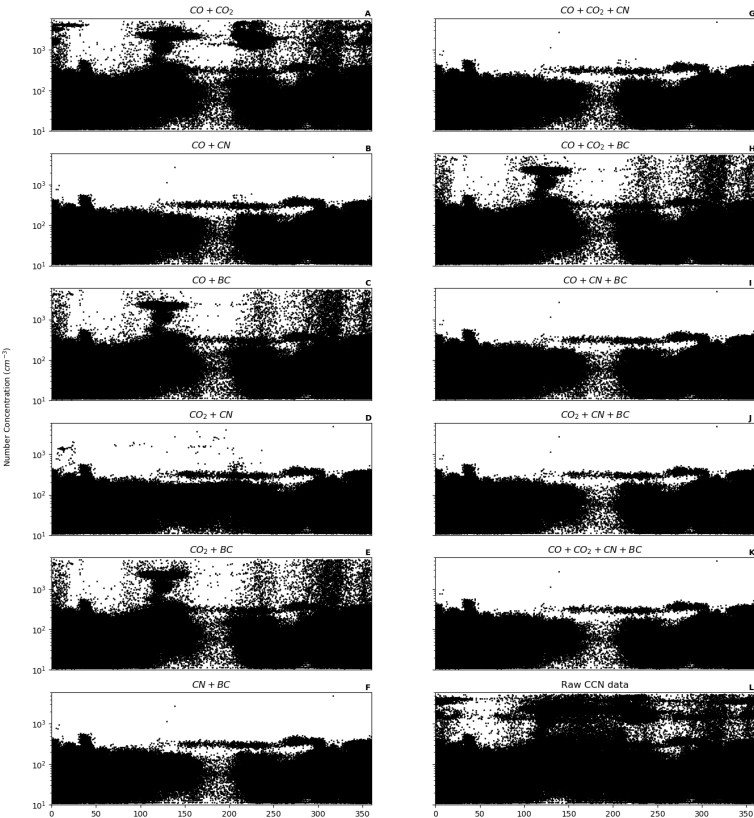

**Figure 5.** CCN data (log scale) with the different combinations of the algorithm applied separately: A-F show all 2 parameter combinations, G-J show 3 parameter combinations, K shows the filter using all 4 parameters, L shows unfiltered data for comparison. Note that the window filter is not applied to any of these plots.

## Appendix A: Outlier Detection Algorithms

Due to the magnitude of differences between the exhaust air and ambient air, exhaust can in the first instance be treated as outliers to the ambient data. The caveat to this is that not-infrequently, the exhaust is itself the dominant influence in the data, making the ambient data itself the outlier. This makes the application of traditional outlier detection algorithms difficult, and is ultimately the reason why a specialised algorithm was developed for operational deployment. During the development stages though, a number of methods were trialled.



Outlier detection methods are classified into six broad groups (Aggarwal, 2013) which include: extreme value analysis, probabilistic and statistical models, linear models, proximity-based models, information theoretic models and high-dimensional outlier detection. Not all of these groups apply to the time series data being considered here.

Fast Fourier Transform (FFT) is a method commonly used to filter outliers from the frequency domain. This is commonly utilised in data that has some level of periodicity or seasonality. Unfortunately, at the short time scales and spatial locations being considered for this application, ambient data do not contain enough periodicity to be able to utilise this method effectively. Nevertheless, an algorithm was trialled which utilised standard FFT functions in Python's NumPy library. Figure A1B shows the effectiveness of the FFT algorithm which was found to be useful for removing some spikes in data caused by exhaust, but struggled during periods of extended exhaust influence.

Generally speaking, environmental data is normally distributed. The distributions of the data can be used to identify an exhaust population, and all the major exhaust influence can be confidently removed using a simple threshold filter. The threshold here becomes very clear when measuring in pristine background conditions, but can become difficult to establish in urban or continental air-masses where ambient and exhaust air compositions converge. FigureA1C shows the data resulting from applying this informed threshold followed by a window filter that identifies data periods within 20 min of an exhaust period as exhaust. Reasonable exhaust removal is achieved compared to other outlier detection methods, however significant exhaust influence remains.

The z-score method is a way of describing data relative to its statistical parameters. In the standard implementation, the z-score of a particular data point is calculated relative to its mean and standard deviation. This obviously has issues if outliers are a dominant feature in a data set since the outliers significantly affect the mean and standard deviation. To improve robustness, the modified z-score compares data to medians and median absolute deviations. In both cases, once the z-score is calculated for each data point, a simple threshold is utilised - that is, if the z score is outside $\pm 3$ ($\pm 3.5$ for modified z-score), the data point is treated as an outlier. In application to this dataset, as shown in Figure A1D, this method functions simply as a threshold filter, removing any data above a certain point, depending on the actual chosen z-score threshold. Applying this method to a rolling window, rather than the full dataset, should improve its performance, however because the rolling z-score calculation tends to simply follow the median of the dataset, its performance actually is not improved.

Double exponential smoothing is a method that creates a model of the data based on exponentially weighted moving averages and linear regression, after which the difference between model and measurements is calculated and compared to a predefined threshold. The method was first described by Holt (2004) in 1957 but with recent advances which included seasonality, became popular in 2000 (Brutlag, 2000) because of its application in time series data for network monitoring. The application of this method here is ineffective in the first instance since it relies on an outlier sensitive method. However, when the model is calculated iteratively on a rolling window, each measurement is determined to be an outlier or not in real-time and replaced, thus increasing substantially the performance of the algorithm (Figure A1E). Despite its impressive performance, a significant influence from exhaust persists in the filtered dataset.





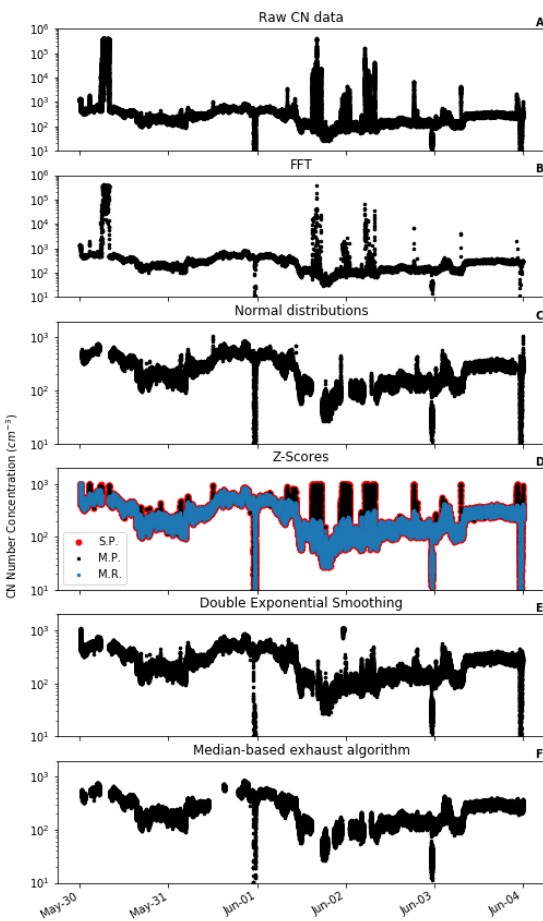

**Figure A1.** A subset of CN during the voyage with a range of outlier detection methods applied. A) Raw CN data. B) Fast-Fourier Transform (FFT). C) Normal distribution filter. D) Z-scores: in red the standard method is applied on the whole population (S.P.); in black, the modified z-score is applied on the whole population (M.P.); in blue, the modified z-score is applied on a rolling window. E) Double exponential smoothing. F) The median based method developed in this manuscript.





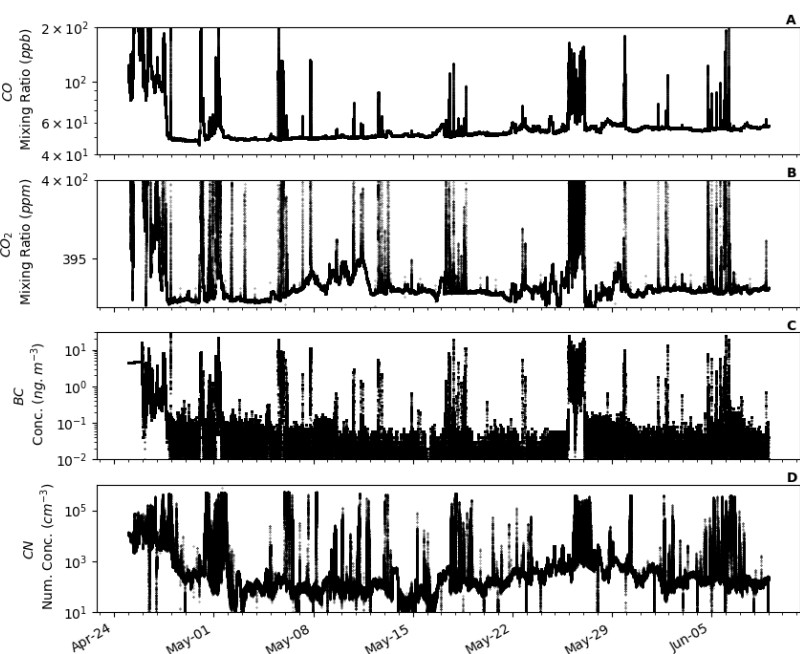

**Figure A2.** Time series (log scale) of CO, $CO_2$, BC and CN for 45 days of voyage IN2016_V03 which traversed from the ice-edge to the equator along the $170^{\circ}$W meridian with a short resupply port period in Wellington, New Zealand on the 26th May, 2016.



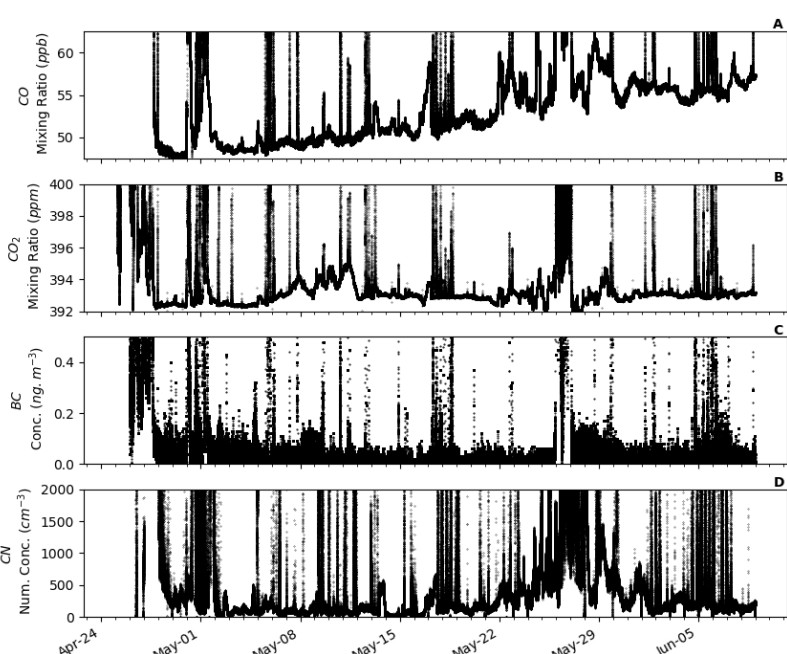

**Figure A3.** As in Figure A2 but with a linear y-scale to reveal the baseline changes.





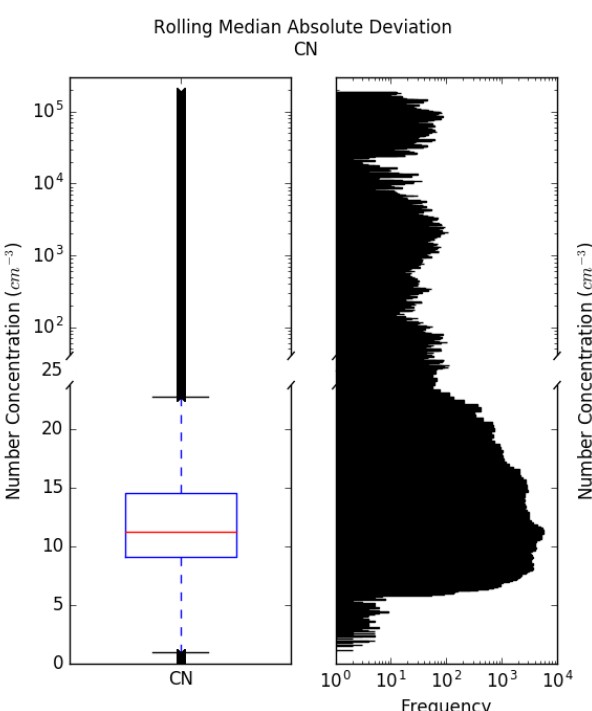

**Figure A4.** Distribution of the MADs calculated from rolling through CN number concentrations. Left: Box and whisker plot with quartiles drawn. Whiskers represent the quartiles $\pm$ 1.5 times the interquartile range. Right: Histogram. Note the split axis which changes from linear to logarithmic scaling.





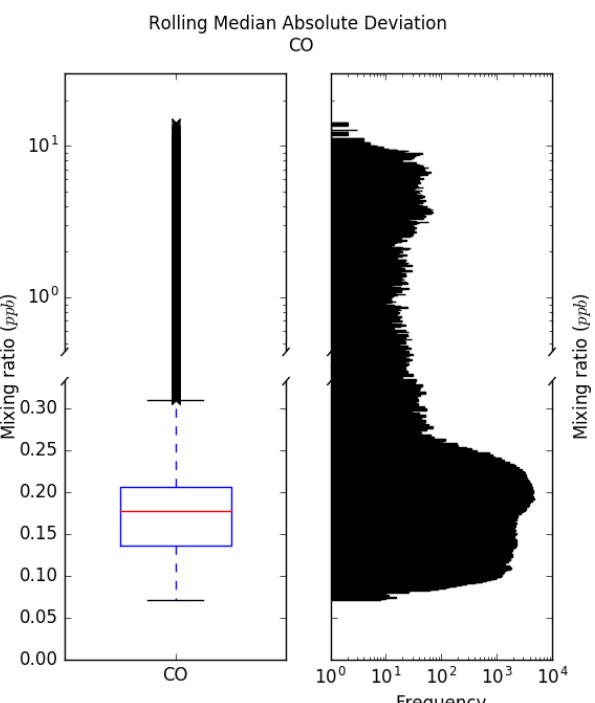

**Figure A5.** As in Figure A4 but for CO mixing ratios.





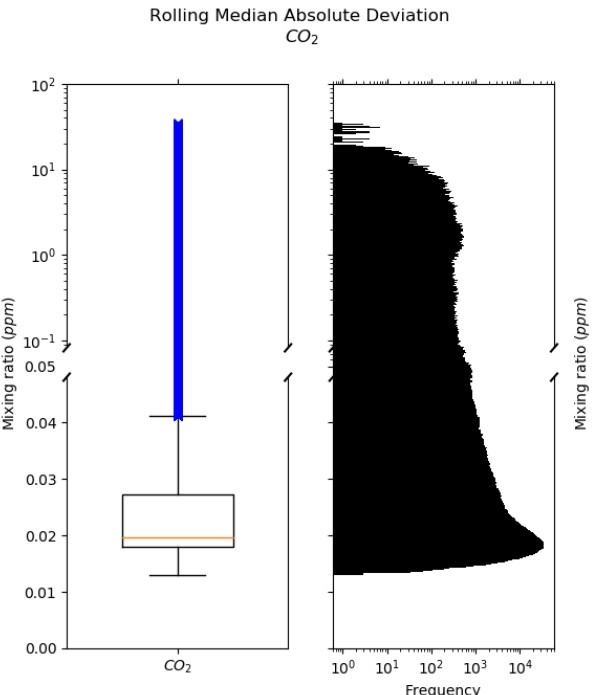

**Figure A6.** As in Figure A4 but for $CO_2$ mixing ratios.



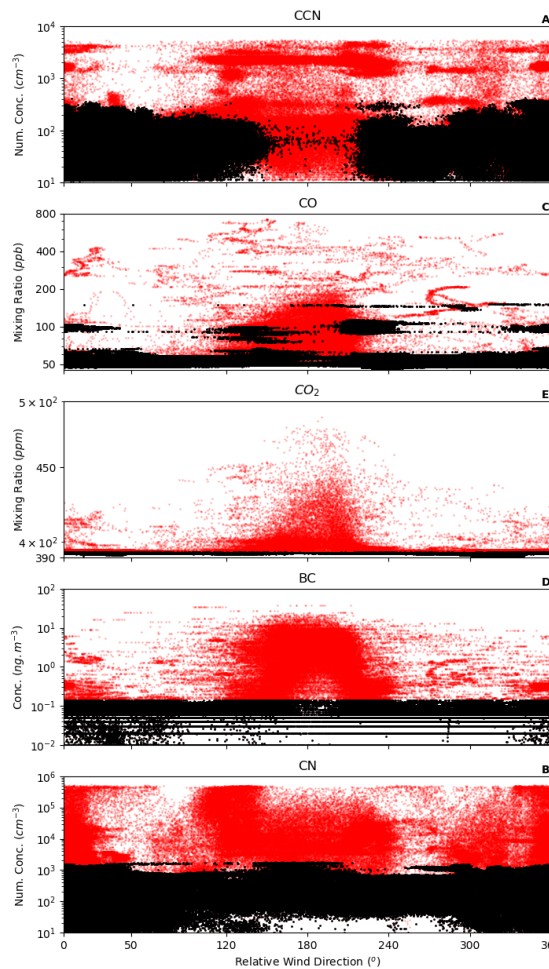

**Figure A7.** As in Figure 3 but plotted against relative wind direction. Unfiltered data are shown in red, while data with the respective exhaust filter are shown in black.





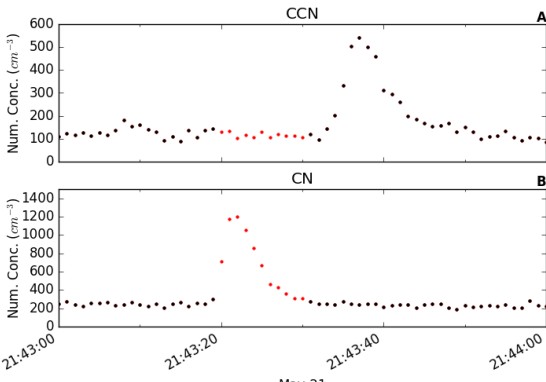

**Figure A8.** Time series of one minute of aerosol data. Unfiltered data in red, with black markers showing exhaust filtered data. The exhaust is clearly identified in the CN data but due to differences in instrument residence time, the exhaust signal shows up 10 seconds later in the CCN data, in this case, after the exhaust signal has ceased in the CN.

*Author contributions.* Humphries developed the algorithm and led the writing of the manuscript. All authors contributed to the writing of the manuscript. McRobert and Ponsonby oversaw the daily instrument maintenance, while Humphries, Krummel, Loh, Keywood and Ward were lead scientists maintaining the calibration and annual maintenance of instrumentation. Harnwell, McRobert and Ponsonby developed and installed much of infrastructure for all instrumentation.

5   *Competing interests.* No competing interests are present.

*Acknowledgements.* The authors would like to thank the Marine National Facility for providing the infrastructure and logistical and financial support for the ongoing measurements on the vessel.



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
