# Peer review of "Identification of platform exhaust on the RV Investigator"

_Atmospheric Measurement Techniques, 2018_

## Referee Comment (RC1) · Anonymous Referee #2 · 13 Oct 2018

Review of "Identification of platform exhaust on the RV Investigator" by Ruhi S. Humphries

General comments:

This manuscript proposed new way for the elimination of self-ship plume using the measured gas and aerosol data. The authors tried to eliminate the relatively high concentration CCN observed in the cruise by the combination of BC, CO, CO2 and CN data. The method is very interesting. However, the premise for target data information in this study is not sufficient. There was less evidence to understand this method advantage from traditional method. Authors described details of this method advantage using several cases by comparison of traditional method. Furthermore, the authors applied the method to only one case by R/V Investigater. This method could be only applied to R/V Investigater.

There are still many discussions to establish this method. I would not recommend publication on AMT in the current form. Significant revisions are needed.

Specific comments:

- Most important point is why the authors mentioned that CCN data captured much number (>1000cm-3) in the relative wind direction within +/- 90 deg from the bow in the Figure1 are recognized to self-ship plume. The comment in Figure1 of "uncontaminated data are usually less than 1000cm-3" is also unclear. First, authors have to indicate that these data are self-ship plume, specifically. Furthermore, why authors have to describe the reasons that these data were captured. It seems that authors just try to reduce high CCN value within +/- 90 deg from the bow.
- In abstract, there is not included the point. Need the quantitative suggestion for the comparison between traditional way and this method. It is should be mention the advantage points from the traditional point.
- 3. For the structure of manuscript, there is unclear information(figures) in the introduction. Their details are important point. However, their explanations were described letter section. Figure1 might be motivation for this study. However, many observation data are included in this figure. Although, this manuscript shows the data analysis method, it makes confusion to read manuscript due to less information for Figure1 (air mass origin, ambient condition, period, location, data sampling time (delta t)). Also, Figure2 is not need in the introduction.
- 4. To discuss the influence the ship plume, authors should describe (or show) the positional relation for inlets, funnel and obstruction in those for the R/V.
- 5. For analysis of data, what kind of data (time step) and residence time in the tube. Also, need the more basic information for the data quality (calibration gas information etc.) in this study.
- 6. This manuscript mentioned new method to the eliminate the self-ship plume by the combination

of BC, CO, CO2 and CN data. Why authors did not use other tracers for ship-plume such as O3 and NO2 ? O3 and NO2 change are also important index for the ship plume. Also, ratios of CO/CO2, CO/BC, CO2/BC are important information.

- 7. Criteria of BC is very sensitive due to low background level. However, evidence for the setting value of 0.07ug/m3 is unclear. Need specifically reason.
- 8. It is afraid that the determined way of criteria of CO, CO2, and CN could not capture the small level event or very large event (not ship plume) in narrow band also will be eliminated. How do you think?
- 9. For window filter setting, this setting is very important. Each observed data would have different time step (1sec or 1min or others?). There is not information for timestep for each measured data in this manuscript. If BC, CO, CO2 and CN have different time steps, the window filter indicate have weighted criteria. Authors have to evaluate this thing (sensitivity check).
- 10. In Figure 3, the BC criteria is 0.07 ug/m3? It looks 0.15 ug/m3 (Figure A7 also)
- 11. In Figure3, it seems high peaks are disappeared from original one. How do you consider these peaks are not from self-ship plume? For example, only CN have high peak (CO, CO2, BC were not high), these cases are real self-ship plume?
- 12. There was not comparison of this method with traditional method. Authors have to show quantitative comparison or improved results. At least, the comparison with the traditional method result has to add the Figure 3.
- 13. In P12 L15, suddenly it starts comparison of wind filter with this method using 42days data. There is also less information for the observed air mass. Although authors mentioned that 20 hours data was recovered (this time might be "net"), these data information are also important. Authors should indicate removed and recovered data property (what is the advantage for this method from traditional one. For example, how percentage recovered data in >±90deg, and removed data in < ±90deg. etc. to compare with traditional one). To indicate this method advantage, authors should compare the traditional method results with this method results in the time-profile figure (like a Figure3). Also, authors should describe the supportive evidence why the data removed or recovered from the data of traditional method using several cases.
- 14. Finally, the figures are not effective show. Please consider more effective show to understand easily. Authors show the many points in the figure. It indicated the data was reduced (or increase) by the filtering. However, there was not information how many points were change by the criteria in the manuscript.

---

## Referee Comment (RC2) · Anonymous Referee #1 · 14 Oct 2018

General comments:

The paper introduces a method on excluding ship exhaust. With this method, the authors expected to identify the periods influenced by ship exhausts and save those exhaust-free data which would be deleted by traditional method (by wind speed and direction). However, the manuscript is not convincing enough to support the conclusion that the method is robust and applicable to other dataset obtained onboard. High uncertainties even misleading might exist. This paper may not be sufficient to be published on AMT. I would suggest a very major revision.

Suggestions for revision:

1. It is necessary to describe main structure of the RV Investigator, especially, providing

the location of the main chimney and its distance to the inlet/aerosol laboratory. This is important to proof that what the authors filtered by the wind direction is mainly from the ship engine exhausts. Also, the authors can give a description of chimney-aerosol inlet distribution in which the method could be useful.

2. Page 3 Line 12, the authors expressed that "not all measured parameters respond to the exhaust simultaneously, or necessarily at all." The explanation following the point is confusing and not really acceptable. The sensitivity or detection limit of the instruments cannot be the reason for not measuring ship exhausts – which is usually shown as extremely high concentration of the tracers like BC, CO, and CO2 etc. The MAAP measured the BC particle with the optical method, so it is hardly missing the ship exhaust particles which is not extremely small (e.g. showing peaks on 40nm and 70nm, see Mar Viana et al., AE, 2014). In Figure 2, only CN showed high value at 0000 on May 19, but the other tracers not. How to proof that the high CN is indeed from the ship emissions? Make sure all other self-contamination sources (e.g. painting on the ship surface, human activities with large emissions) could be excluded.

3. Regarding BC threshold filter, why chose 70ng/m3 as the threshold; why not using the rolling window method as CO, CO2 and CN?

4. The rolling window with waving criteria for filtering data can eventually exclude the outliers. This could be a better way than the constant criteria and can save a lot of data. However, the sources of the outliers have to be clarified before deleting the data. Similar to the 2nd point, how to make sure the extremely high values around 0000 of May 19 do represent the ship exhausts while BC, CO, CO2 cannot say this. The present method excluded also the high values out of the range 90° to 270° of the relative wind direction, are these points corresponding to the data on 0000 May 19? If yes, does this mean these high value points may not be related to the exhausts?

5. Also, the authors recognized that CN would be the most useful parameter for data cleaning and took the CCN as an example. Since only CCN was tested, this conclusion might only work for CCN. How is the situation of aerosol components like sulfate, organics, or particles in small size range (e.g. 40nm which are usually the size for fresh engine exhausts)? The authors may want to give more examples on other chemical compositions, or even applications on other datasets to show this method can work universally.

———————————————

---

## Author Comment (AC1) · 11 Dec 2018

We would like to thank the referee for their insightful comments. By responding to the referee's comments in the revised manuscript, we expect the manuscript to be significantly improved. In responding to the referee's concerns, we will respond to each comment specifically.

1. We agree that an overview schematic of the ship will be useful in the presentation of the manuscript, and this will be included in the revised version. If the referee could provide clarification of the "chimney-aerosol inlet distribution" comment, we would appreciate it.

2. While we understand the referee's concerns about the lack of signal in some of the

parameters, we respectfully disagree with their conclusions. While all the parameters will respond equally well to very strong exhaust strikes, their response differs to minor exhaust strikes. The referee particularly notes the MAAP using the optical method, however we note that the specification of the MAAP states clearly the minimum detection limit of a 2 minute average of data is 100 ng/m3. Since we are utilising 1 Hz data, it is quite likely that weaker exhaust signals just won't be seen above the noise of this instrument. We will undertake a case study of the exhaust period around 0000 on May 19 in the revised manuscript to try to investigate why the other tracers don't response, and rule out any other sources of strong aerosol particle counts.

3. The choice of the 70 ng/m3 threshold, as well as the choice not to use a rolling window method is discussed quite clearly in section 3.1. Consequently, no changes will be made in the revised manuscript.

4. If the referee could provide a reference describing the "rolling window with waving criteria" method for outlier identification, we would be happy to test this method as we have tested a number of other methods. The case study that will be undertaken for the revised manuscript in response to the referee's second point will address their other concerns in this point.

5. The referee's comments here are quite valid, and this could be the case. We had chosen CCN as it was readily available for that voyage, is measured at the same high frequency as the exhaust was calculated (1 Hz) and has a clear exhaust signal. We also had real-time aerosol composition and aerosol size distributions that were measured during this voyage, which were measured at lower time resolutions. We agree with the referee that the showing the algorithm's application to other datasets will dramatically increase the evidence for its robustness and so we will apply the filter to these datasets in the revised manuscript.

---

## Author Comment (AC2) · 11 Dec 2018

We would like to thank the referee for their review of the manuscript. In addressing the points, we expect a significant improvement of the manuscript. In responding to the referee's concerns, we will respond to each comment specifically. 0. The referee's general comments were primarily concerned with the applicability of the method beyond its use on the RV Investigator. We note that the method being described has been developed specifically for the RV Investigator platform and its use beyond this platform, while possible, is not within the scope of this manuscript. The manuscript serves primarily as a description of the algorithm utilised on this platform only. 1. The data utilised in the manuscript are from periods in the remote marine boundary layer of the Southern and Pacific Oceans. CCN concentrations in these regions are known

to be very low, in the concentration range of tens to a few hundred. Concentrations above this are not observed, and those above 1000 cm-3 are rare even in urban air in the Southern Hemisphere. In addition, when looking at the time-series of CCN, CN, BC, CO, CO2 and relative wind direction, it is clear that these periods are from local ship exhaust rather than any natural phenomenon. We will include a figure in the appendices to this show this clearly in the revised manuscript. In addition, the goal of algorithm is not to reduce the high CCN values, as the referee suggest. The use of CCN was intended purely as an effective and obvious indicator of exhaust due to the relatively low background concentrations and high exhaust concentrations, but without being part of the algorithm itself.

2. We agree that a quantification of the advantage of this method over the traditional methods should be included in the manuscript, and in particular in the abstract. We will do this in the revised manuscript.

3. Figure 1 includes all the data from the full voyage data undertaken. While we haven't included the other parameters mentioned by the referee in the figure, we note that the goal of the algorithm is to work as a near-real-time algorithm where much of that information is unavailable, so inclusion of that information in Figure 1 isn't really applicable. However for the reader to properly understand this dataset, we will include a map of the voyage plotted in Figure 1 as one of the appendix figures. Figure 2 is referred to in multiple locations within the manuscript. In the introduction, it is intended to show the reader what the raw time-series looks like, which is essentially what the algorithm is processing. It shows the distinctly different signals in each of the parameters included in the algorithm, which is important in giving context in the introduction.

4. We agree with this comment and will include a schematic overview of the RV Investigator for the manuscript.

5. Residence times – due to the way the algorithm removes data within a 20 minute window of a positively identified exhaust period, the question of residence time is not

important for the manuscript, and so has not been included. We will clarify this more explicitly in the text. Calibrations – we have described in the manuscript why we use uncalibrated data in the algorithm – this is primarily because we are looking for relative changes in data signals, and calibrations do not effect short-term relative changes. This has already been dealt with in the manuscript.

6. The choice of tracers utilised by the algorithm were based on data availability and its effectiveness in the method. $O_3$ was tested and found not to be effective, presumably because of the timescale of the chemistry involved, or the fact that the ship burns cleanly relative to other ships where $O_3$ is a good tracer. $NO_2$ is not part of the permanent instrument suite aboard the RV Investigator, so is unavailable for use. The use of ratios is not likely to extract any additional information for the purposes of exhaust identification other than what the individual data streams utilise.

7. BC is indeed the common tracer for exhaust. However, because of the noise in the data at the high frequency being utilised (1 Hz), it wasn't as useful as expected. The manufacturer of this instrument recommends averaging 1 Hz data to 20 min time steps in order to get effective signal-to-noise ratios. The chosen value of 0.07 ug/m3 hasn't been described as to why this was chosen, and this will be included in the revised manuscript .

8. The criteria utilised in the algorithm was chosen so as to distinguish between self-ship exhaust, and the background atmosphere, which includes polluted urban environments where the ship might port. If a signal comes from a nearby point source (i.e. a close-passing ship), the algorithm will likely incorrectly identify this as self-ship exhaust. For small-scale events, the sensitivity of the algorithm can be tuned by the user in order to identify, or not, that event, at the cost of falsely identifying non-exhaust data. In any case, the algorithm is not an end-to-end solution, and the data must be examined by human eye before a final dataset is published and applied to other datasets.

9. All the data utilised in this algorithm are 1 Hz data. This has been included explicitly

for some instruments, but not for all, and will therefore be stated more clearly in the revised manuscript . Because all the data input are at the same frequency, the window filter weighting isn't a concern.

10. The referee has a keen eye! We will ensure to double check this in the revised manuscript .

11. The disappearance of peaks between Figure 2 and Figure 3 is likely a result of differences in plotting (i.e. changes in colours and marker sizes). We assure the referee that the underlying data is the same. Nevertheless, we will replot these figures ensuring consistent plotting parameters between the figures . In regards to why an exhaust signature is visible in some parameters but not others, we believe this is a result of the sensitivity and response times of the particular instruments, with CN being the most sensitive to this particular exhaust signal.

12. We will already include a quantification of this method in the manuscript as per the referee's previous comments. We will consider adding the traditional algorithm to Figure 3, or at the very least, will add an additional Figure in the appendices.

13. Our previous amendments to the manuscript from points 2 and 12 above cover most of the concern here. The addition of case studies to illustrate why the new algorithm recovers or removes data compared to the traditional filter will be a useful addition, and will be included in the revised manuscript.

14. The amendments that address points 2, 12 and 13 above will address the reviews concerns in this point.

---

## Author Response (AR1)

Author's record of changes made to manuscript AMT-2018-214 after peer-review:

In response to Anonymous Referee 2 (RC1) comments:

0. Point 0 – as addressed in the previous author's response. No changes were made.
1. Point 1 – a figure has been added to the appendices, with reference from the caption of Figure 1, showing why the 1000 cm-3 is a reasonable limit.
2. Point 2 – Quantification of this method has been performed and the results have been included in the abstract as well as in the final paragraph of the results section.
3. Point 3 – Figure 2 has been removed, as suggested by the reviewer, and reference to the next figure has been used to illustrate the same points. In addition, a voyage overview map has been added as a figure in the appendices to give the reader greater context.
4. Point 4 and RC2, point 1 – a schematic overview of the RV Investigator has been added to the appendices.
5. Point 5 – additional text has been added to the "Window Filter" section to further clarify the point that the window filter covers any residence time differences.
6. Point 6 – as addressed in the previous author's response. No changes were made.
7. Point 7 – the section describing the BC threshold filter and the choice of the 0.07 ug/m3 threshold is very clear already, and further clarification of how the threshold was chosen is unnecessary. No changes were made.
8. Point 8 – as addressed in the previous author's response. No changes were made.
9. Point 9 – the timebase of all the data has been clarified in the "Instrumentation" and "Exhaust Identification" sections.
10. RC1 point 10 – the data was verified and the exhaust algorithm was rerun and all plots reproduced. This fixed the issue with the BC plots not reflecting what was in the text, and another time offset was discovered in the aerosol concentration data that was corrected, which resulted in changes to other graphs, but no changes to the results.
11. Point 11 – all data has been replotted after recalculation of the exhaust algorithm and plotting routines have been checked to ensure the same data is plotted consistently with the same marker sizes and colours.
12. Point 12 – we have added quantification of the method compared to the traditional algorithm in the text, but have also added additional figures in the main text, and in the appendices, comparing the wind-based filter to the algorithm.
13. Point 13 – additional figures have been added in response to point 12, which directly address this point. The final paragraph of the "Results and Discussion" section has also been edited to discuss this idea further.
14. Point 14 – as discussed in the author's response to the discussion, amendments that address points 2, 12 and 13 of RC1's review address the concerns for this point.

In response to Anonymous Referee 1 (RC2) comments:

1. Point 1 – A schematic overview of the ship has been added to the manuscript.
2. Point 2 – we have found a time-offset in the aerosol concentration measurements which has been fixed and the algorithm rerun. Although this removes the misalignment of some of the exhaust strikes, there are still some smaller exhaust strikes which show signal in CN, but not in the other parameters. The reasons for this is the magnitude differences of the exhaust signal in each parameter, as well as the sensitivity of each instrument. Exhaust shows up in CN data as changes of up to 4 orders of magnitude, while BC, CO and CO2 change by factors of only 10, 0.2 and 0.01. So a small exhaust signal might show up as a small signal in CN, and

nothing at all in the other parameters. The 6th paragraph of the introduction has been edited to reflect this discussion, as well as an additional figure added to the appendices.

3. Point 3 – as addressed in the previous author's response. No changes were made.
4. Point 4 – as addressed in the previous author's response. No changes were made.
5. Point 5 – unfortunately no data is available for aerosol composition in the size range of interest for the main exhaust particles (sub-100 nm), so comparison to these data were not possible. However, the application of the algorithm has been applied to aerosol size distribution from an SMPS, and a paragraph added to the results and discussion section, as well as a Figure in the appendices.

[revised manuscript text omitted]

---

## Author Response (AR3)

**Decision and comments from the Associate Editor, with responses by authors inline in green text.**

11 April 2019

**Associate Editor Decision: Publish subject to minor revisions (review by editor)**
(10 Apr 2019) by Wiebke Frey

Comments to the Author:
Dear Ruhi Humphries et al.,
thanks for the revision of your manuscript.
There are a few points that should be addressed before publication, please see below.
Best wishes,
Wiebke Frey

**Major comments:**

One of the reviewers still feels that you do not address the advantage of this new method over other methods enough. You do mention some advantages (e.g. minimizing false positives) in places, but I think it should be possible to add some more text to discuss and make the advantage of this method more clear. Please also add text to the conclusions, where the advantage is not mentioned at all.
We have assessed the manuscript and found the advantages of this method are already discussed in the Abstract and Sections 1, 3 and 4. We have made slight amendments to Section 4, and more substantial amendments to the conclusions section to explicitly discuss the advantages of this algorithm.

**Minor comments:**

page 5, line 8-9Can you be more specific about the 1 Hz signal detection limit ("much higher")?
We have calculated the detection limit for the dataset via the 3 x standard deviation method on a "blank" period where no BC sources exist (i.e. the deep southern ocean). This detection limit was below larger than those stated in the instrument manual for 10 and 20 minute periods, but lower than the chosen threshold. We have amended the text in Sections 2.3 and 3.1.

**Figures:**

A1 - Please indicate the periods of elevated CCN concentrations across all panels (vertical colour shadings), mention the according range of wind directions in the figure caption. This will guide the reader to the parts where data are affected.
The reviewer's request seems to be referring to the idea that exhaust signal observed in the CCN data typically pushes concentrations above 1000 $cm^{-3}$. This threshold is a guide, rather than a rule. Applying vertical colour shadings to the graph with this threshold will be rather arbitrary, as it doesn't correspond to all the exhaust signal, and will misguide the reader. Similarly, if vertical shadings were added when exhaust was detected, the graph would be unreadable. The purpose of this graph is simply to show what the raw time-series looks like and how similar signals exist in multiple datasets. We have edited the caption of this Figure to clarify that the 1000 $cm^{-3}$ is a guide, rather than a rule. No changed to the actual figure have been made.

A2: Check references to Figures, Fig. A3 (and A4, A5) are mentioned in the text before A2. Maybe a

reference is missing?

We have reordered the Appendix figures so that this skip in numbering in the main text does not occur. Specifically, the appendix figure previously labelled A2 (which was the figure associated with Appendix A text), has been moved to the end of the Appendix figures, and is now labelled A14. Figures previously labelled A3 to A14, have been renumbered as A2 to A13, respectively.

A3: It would be helpful if you could additionally repeat the point you want to make with the figure in the figure caption (add information to the figure caption).

The caption for the figure (now Figure A2) has been amended to include more information – specifically, the information regarding why this particular dataset was chosen.

A10 and according text in manuscript: Are the instrument data time synchronised? It does not look like! If synchronised, those peaks would presumably fall on the same time. Such systematic time shifts between instruments should be identified and synchronisation applied before filtering!?!

We presume the reviewer is referring to Figure A11 (which after the reordering of Appendix figures is A10) – the time series of one minute of aerosol data. All these data are aligned on the time dimension prior to the final exhaust window being applied. As discussed in the Figure caption and the text, this is likely a result of differences in residence times in the instrument. We have added text to Section 3, 3.3 and 4 to explicitly state that the different data streams are aligned on the time dimension.

[revised manuscript text omitted]